# Final Conversations: Overview and Practical Implications for Patients, Families, and Healthcare Workers

**DOI:** 10.3390/bs7020017

**Published:** 2017-04-05

**Authors:** Maureen P. Keeley, Mark A. Generous

**Affiliations:** 1Department of Communication Studies, Texas State University, 601 University Drive, San Marcos, TX 78666, USA; 2Department of Communication, Saint Mary’s College of California, 1928 St. Mary's Road, Moraga, CA 94575, USA; mag31@stmarys-ca.edu

**Keywords:** final conversations, end-of-life communication, family communication, death and dying

## Abstract

The current paper presents a summary of a 12-year body of research on final conversations, which will be useful for healthcare providers who work with patients and family nearing the end-of-life, as well as for patients and their family members. Final conversations encompass any and all conversations that occur between individuals with a terminal diagnosis and their family members (all participants are aware that their loved one is in the midst of the death journey). Final conversations take the family member’s perspective and highlights what are their memorable messages with the terminally ill loved one. In this paper the authors highlight the message themes present at the end-of-life for both adults and children, the functions each message theme serves for family members, and lastly, the communicative challenges of final conversations. Additionally, the authors discuss the current nature and future of final conversations research, with special attention paid to practical implications for healthcare providers, patients, and family members; also, scholarly challenges and future research endeavors are explored.

## 1. Introduction

Close relationships are critical to the end-of-life (EOL) journey, both for terminally ill people and their family members [1]. In particular, it is the communication between these two parties at the end-of-life that has been shown to have a profound impact on the EOL journey [1,2]. Here, the focus will be one aspect of EOL communication known as final conversations, which include all interactions, verbal and nonverbal, that an individual has with another who is terminally ill from the moment of a terminal diagnosis to the point of death [1]. Final conversations may involve only one conversation, but they can also be (and often are) a series of conversations [2]. Final conversations research has focused on the themes, functions and impacts of communication at the end-of-life from the perspective of the family members, close friends, and other individuals that are allowed within the inner sanctum of a terminally ill person’s life, which until recently, was an understudied perspective in the EOL literature [2]. The current manuscript seeks to summarize this body of research, while also addressing three primary insights gleaned from this work: practical implications for healthcare providers, patients, and families; scholarly challenges; and, future research directions. 

## 2. Summary of Final Conversations Research

Prior to 2004, scholars exploring familial interactions in the midst of death and dying primarily focused on the perspective of the terminally ill person [2]. It is only within the last fourteen years that a focus on the “other” person (i.e., family members or close others), through the participation in final conversations, has been explored [3]. Findings suggest that final conversations have the potential to have a tremendous impact on family relationships (biological, legal, or chosen families) [1,4]. Specifically, the realization that time is limited because of a terminal illness increases the urgency for both the terminally ill person and family members to say final goodbyes, and to try to make amends if necessary in their relationships; it also creates a window of opportunity for people to make time in their busy lives to focus on the relationship with the terminally ill person through their participation in final conversations [1]. Final conversations often give terminally ill people the opportunity to help their family members move forward after the death by providing advice, direction and permission to move on, as well as creating a sense of closure and completion of the relationship [1,2]. It is only through communication that terminally ill people and their family members can work together to achieve greater meaning about life, death, and their relationships [3].

### 2.1. A Brief Overview of Methodology

The final conversations body of research uses both qualitative and quantitative methodologies. The data was collected in three phases over a fourteen year period. Specifically, Phase I and Phase II of data collection consisted of in-depth retrospective interviews with adults (Phase I) and children/adolescents (Phase II) [2,4]. Retrospective accounts have proven to be especially clear for family members reflecting back on their EOL communication with the terminally ill person [2,4]. A sampling of the questions of the interviews that led to the most substantive findings include: “Would you share with me your recollection of your final conversation or conversations with your loved one?”; “What was the most meaningful conversation that you had with this person?”; “Why was it the most meaningful to you?”; “What sorts of nonverbal experiences stick out in your mind from this time period?”; and, “What did each of these nonverbal experiences mean to you?” The language of the questions was appropriately adapted according to the age of the participants. Phase I had a total of 85 adults (age range: 21–85) and Phase II had a total of 65 children/adolescents (age range: 5–18). Interviews were collected until saturation was reached, meaning that no new information was being revealed. An interpretive paradigm was used to analyze the data to ensure that the themes captured the authentic and significant experiences of EOL communication from the family members’ perspectives, which facilitated the categories to emerge from the data. 

Phase III employed a cross-sectional survey design in order to create and test initial validation of a Final Conversations Scale [5]. The scale was developed based on past literature, as well as findings from the previous qualitative phases (see [5] for the scale and findings). One-hundred fifty-two participants completed the survey by retrospectively recalling their final conversations. 

### 2.2. Final Conversations Themes and Functions

Five overarching themes emerged from Phase I and Phase II: love, identity, religious/spiritual messages, everyday talk, and difficult relationship talk [2,4]. These five themes—with the exception of difficult relationship talk, which was not found in a children/adolescent sample—emerged in both adult and children/adolescent samples. These messages include both verbal and nonverbal messages [2]. Nonverbal communication is a critical aspect of final conversations, because as death nears, the ability to talk and verbalize words often becomes more challenging and limited for the terminally ill person [6]. Not all themes are present in every final conversation, although often two or three themes can exist within the same conversation; at the same time, in some relationships, one theme will take precedence and be the focal point of the entire conversation, which highlights the importance of contextual elements on the enactment of final conversations [1]. 

The first theme—messages of love—is the most prominent theme for adults to communicate with the terminally ill family member at the end-of-life [7]. Love is communicated both verbally through words, as well as nonverbally via hugs, looks, hand holds, kisses, or other expressions of love [1]. There is not a right or wrong way to communicate love, it is simply important for individuals to communicate it in a way that the other person will understand. Adults and children/adolescents note that messages of love helped to validate and strengthen relational bonds with the terminally ill family member [2,4]. 

The second theme is verbal messages related to individual and relational identity [2]. Identity messages signify the statements that represent the assessment or formation of the self [8]. These messages may contain new information for the family member (e.g., advice, messages of insight and confirmation) [1], or they may highlight known (but perhaps previously downplayed) attributes [2]. When faced with the impending death of a loved one, individuals often take the opportunity to examine, reevaluate, and even redefine themselves because of their final conversations, which was a common function of identity-related messages articulated by participants [2]. 

The third theme centered on religious/spiritual messages, which may be direct affirmations of their faith or spiritual experiences [9,10]. Religious messages often incorporated doctrinal and denominational experiences that include specific behaviors, beliefs, or rituals of a system of worship from a specific religious group that the terminally ill person or family members identified with or shared [9,10]. Spiritual experiences are a phenomenon described as a transcendent occurrence that has deep meaning for the individuals and greatly impacts their belief in an afterlife [9,10]. These spiritual experiences might be encountered by the terminally ill person or a family member prior to their final conversations, or during the EOL journey. Religious/spiritual messages functionally help validate individuals’ beliefs in a higher power and their expectation that they will meet again someday in heaven or whatever comes after this life [9]. In addition, religious/spiritual messages provide comfort and solace for terminally ill people and family members during an often chaotic, uncertainty-inducing time [2,9,10].

The fourth theme, everyday talk and routine interactions, focuses on the ordinary, commonplace conversations and repeated types of daily interactions (e.g., discussing daily activities, talking about television and movies, reminiscing, sharing stories, etc.) [1,2,4]. This was the most prominent theme that emerged from children/adolescents interviews about final conversations [4]. Everyday communication performs numerous and simultaneous purposes within families, including: building bonds, coordinating interactions, structuring time and sharing histories [2]. At the end-of-life, all of these functions are critical for the family members, as they give a sense of normalcy and control in the midst of chaos that an impending death often brings to the situation [4]. 

The fifth memorable theme is difficult relationship talk, which often includes an account of the challenges in the relationship between the terminally ill person and the family member prior to the illness, as well as the struggle that the family member had in talking with the dying person [2]. Difficult relationship talk includes messages that revealed attempts at understanding, accepting, or beginning the forgiveness process towards the terminally ill person [2]. For the individuals that have these challenging relationships, they often report that difficult relationship talk was the most important to them during their final conversations. Additionally, family members highlight that they have to find the right time and the resolve to engage in the conversation, and they hope to finally have a positive interaction with the dying loved one before death. Individuals are hoping to find a way to release pent up anger and frustrations so as not to be left holding these negative emotions following the death [2]. At the same time, some individuals also talk about avoiding certain conversations so as not to make their relationship worse [11]. It is also important to note that interviews with children and adolescents did not uncover the theme of difficult relationship talk. Perhaps the absence of difficult relationship talk from children/adolescent’s interviews is because they don’t realize how challenging the relationships are yet, or because they don’t have the cognitive and communicative ability to participate in difficult relationship talk [4]. 

We were interested in quantitatively validating the aforementioned themes via the construction of the Final Conversations Scale [5]. While constructing the scale, the a priori decision was made to include scale items that measured instrumental talk, which includes discussions regarding the death and dying process (e.g., discussions about the illness, funerals, chores to be completed after the death, etc.). Although this theme did not emerge during the qualitative interviews, we argued that this theme is an important, yet potentially neglected topic of conversation during the EOL journey [3]. A possible explanation for why instrumental talk did not emerge during in-depth interviews is because participants did not recall those conversations as memorable or significant to the relationship. Previous research examining EOL communication highlights the significance of this topic for terminally ill people and reveals that instrumental death talk is occurring [12], yet may not be the most memorable messages for the family members upon reflection and recall. Instrumental talk messages are functionally important, as they can help family members and patients discuss and negotiate needs, desires, and advanced directives [13]. Analyses of the quantitative survey conducted in Phase III revealed a five-factor scale that included messages of love, spiritual/religious messages, difficult relationship talk, everyday talk, and instrumental death talk [5]. Due to possible inadequate operationalization, identity did not emerge as a meaningful factor in the quantitative analyses; however, we are currently working to revise the operationalization of identity, as well as replicate and confirm the aforementioned themes. Specifically, we returned to the qualitative analyses to create more behaviorally concrete scale items to measure the identity dimension of the Final Conversations Scale.

Overall, Keeley and her associates’ findings revealed that communication at the end-of-life is as important for the family members as it is for those that are dying [1,2,3]. This conclusion however should not imply that participating in final conversations is an easy task. On the contrary, communicating in the midst of grief, fear, and uncertainty can be an overwhelming endeavor for both the terminally ill person and family members [1]. High emotions frequently complicate and often interfere with effective communication [1]. In addition, many individuals have never seen or been a part of final conversations and feel ill-equipped for communication at the end-of-life [1,2]. Some of the communication issues include: how to begin the conversations, what topics should and should not be talked about at the end-of-life, who is in charge of leading the conversation, and how much emotion can be displayed during the interaction [1]. Consequently, family members can find the task daunting with many challenges to overcome [14], or they simply choose to avoid certain topics altogether [11]. 

### 2.3. Challenges

Due to the fact that final conversations usually occur in private, and most individuals have little to no experience with communication at the end-of-life, people are uncertain about the right timing for these important conversations [2]. In addition, family members often face certain tensions and apprehensions about what and how to talk with the terminally ill person [14]. The following section summarizes the communicative challenges and difficulties noted by individuals who have engaged in final conversations. 

The first challenge, which also influences the manifestation of other communicative difficulties, is time; specifically, in previous articles we discuss the issue of terminal time, which we define as all moments that occur between the terminal diagnosis and death of the terminally ill person [1,14]. Time creates both the impetus for final conversations, as well as the framework for the communicative context at the end-of-life [1,14]. The framework for EOL communication comes from terminal time which constructs a structure, agenda, and background that are fundamental parts of final conversations. At the end-of-life, time creates a structure for the conversations in that there are hard boundaries surrounding the conversations because there is a beginning (i.e., diagnosis of terminal illness) and an end (i.e., death) regarding the availability for the conversation. It also can create an agenda for the conversations (i.e., What do I want or need to say to the terminally ill person before their death?). Lastly, it establishes the background for every conversation (i.e., How is the terminally ill person emotionally feeling and physically looking today? Are they in pain? Do they have enough energy to participate in a conversation? Am I ready for this conversation?) [1]. For some, the diagnosis of a terminal illness creates urgency and awareness that time may be running out for opportunities to communicate and interact with the terminally ill person [14]. Unfortunately for others, they wait too long to participate in final conversations. As death nears, terminally ill people often suffer from extreme physical fatigue, difficulty in speaking and even mental deterioration, which can make it nearly impossible to have any substantial communication with them during the later stages of the dying process [12]. Thus, the physical and mental state of the terminally ill person may provide a tangible barrier to effective, open communication [11]. 

The remaining challenges related to final conversations can be understood via relational dialectics theory [15]. Dialectical tensions refer to the “dynamic interplay of opposing forces or contradictions”, which are illustrated by dialogue between relational partners [15] (p. 3). Three dialectical tensions relevant at the end-of-life include: the acceptance-denial of the impending death; openness-closedness regarding how honest and revealing they are with each other on a wide variety of topics; and, the expression-concealment of emotion that occurs during the death journey [14].

The tension of acceptance-denial highlights the struggle that many individuals deal with when faced with the news that their loved one has a terminal illness [14]. Some individuals refuse to accept the terminal diagnosis because it causes them anguish and anxiety [16]. For others, it may simply be that they are afraid that if they accept the diagnosis, that they are giving up hope, and in part, inducing a faster death [14]. By accepting the impending death, individuals are able to face the truth of the situation, thereby allowing them to make the most of the time that they have left with their dying loved one [1]. 

The tension of openness-closedness represents individuals’ struggle with the desire to self-disclose information to the terminally ill person, but also to keep some information private [14]. For instance, some families have norms regarding what should and should not be talked about based on their family history, such as personal and private information about themselves, or negative relationship issues. This avoidance may be a way to manage potentially negative emotional responses (i.e., self-protection and other-protection) [11].

In addition, some people worry that various topics could upset the terminally ill person, therefore causing unnecessary burden [11,14]. In fact, both family members and the terminally ill individuals may avoid topics and act positive in the midst of the death journey as a way to protect each other, because they believe the other to be too vulnerable to have honest and open conversations about death and dying [17]. Even more problematic is that sometimes both the terminally ill person and family members mistakenly think the other person doesn’t want to talk about the difficult experience of death and dying, when in reality they do, which leads to a missed opportunity for dialogue and connection [18].

The tension of expression-concealment of emotions highlights the struggle to show the emotions they are experiencing juxtaposed with the desire or expectation to be strong and hide their true feelings at the end-of-life [11]. Strong negative emotions such as sadness, fear, or anger are inevitable at the end-of-life and most people are not good at expressing negative emotions [19]. From childhood through adulthood, many of the responses to the expression of negative emotions are gender based. For instance, boys are socialized not to show sadness by telling them “big boys don’t cry”, or they are congratulated for “being a strong and brave little man” [19]. Women are chastised for showing anger and are often called negative names for displaying their anger; this is so common that many women report trouble communicating their anger and instead they cry [19]. Culture also plays a big role in the display of all emotions, especially negative emotions because they are a bigger threat to the positive image of the individual or the larger community depending on the values of the society [19]. Finally, some families have expectations that they are supposed to be strong for one another and not display emotions related to sadness, anxiety, and distress [11,14].

## 3. Practical Implications

The research summarized above has implications for healthcare and palliative care providers, as well as the terminally ill person and their family members. First, final conversations research has helped us understand the interpersonal scripts between terminally ill patients and their family members that accompany the EOL journey [2]. Interpersonal scripts are important to consider in any communicative context, as they represent working models of individuals’ communication behaviors and choices [20]; but, this concept is especially poignant in the context of family communication at the end-of-life. The US, and Western culture broadly, tends to view death as uncertainty-inducing and scary, which could potentially lead to an underdeveloped interpersonal script and avoidance surrounding the EOL context [11]. If, however, people become more aware of final conversations, see examples of what they look like, and begin developing their own scripts concerning the end-of-life, then people will potentially begin the conversations sooner with more fulfilling outcomes. Why is this? Because currently, many family members wait until the very end to have these conversations, and by then it is too late because terminally ill people are in the active stages of dying, where they are often not physically capable of verbal interactions [6].

Second, raising family members’ awareness about final conversations can help facilitate dialogue regarding needs and desires of the terminally ill person and family members related to EOL communication. Providing examples and giving encouragement to participate in final conversations earlier in the death journey may enable participants to have a better EOL experience. Participating in communication at the end-of-life could also help decrease individuals’ fear of the death process and help to change the culture of silence and uncertainty surrounding death [1]. 

We have argued that individual needs regarding communication at the end-of-life vary from person to person and from context to context [3]. It is the task of healthcare and palliative care professionals to use the tools available to them to help families articulate their communicative needs. For instance, the Final Conversations Scale is available, which is a measure of EOL relational communication that assesses verbal and nonverbal messages that occur at the end-of-life (see [5] for scale). Although this scale was originally designed to assess retrospective accounts of EOL communication for research goals, it can be adapted as a checklist to capture communicative needs of family members currently engaged in final conversations, as well as to help generate new scripts for both the terminally ill person and their family members. For example, palliative care professionals can adapt the scale’s wording to present tense and change the scale’s numeric anchors to assess needs (e.g., 1 = I do not want to talk about this; 7 = I want to talk about this). Furthermore, dialoguing with terminally ill people and their family members can help healthcare providers understand the needs, fears, and desires concerning their EOL journey. 

Finally, the issue of children and adolescents’ engagement in final conversations should be addressed with regard to practical implications, as this population is frequently shielded from the death and dying process by family members in an effort to protect them [3]. We analyzed messages of advice from children/adolescents to other children/adolescents and adults regarding final conversations [21]. (Mainly, these participants advised individuals to focus on confirming the relationship between the terminally ill patient and child/adolescent, as well as family members talking sooner rather than later with the children/adolescents [21]. In addition, these conversations need to be candid, open and honest regarding the patient’s status and progress. Children/adolescents are not clueless and can often see that something serious is occurring between family members. Thus, by talking sooner rather than later, children/adolescents are given the chance to participate in final conversations on their own terms and in their own time. From the interviews we discovered a slight paradox—family members tend to believe that children/adolescents are too young and fragile to handle final conversations and death, but the children/adolescents desire more transparency from family members [21]. Thus, we advocate for a collaborative communicative approach to talking with children and adolescents about final conversations and death: Ask, Look, and Listen (ALL) [21]. In particular, family members should Ask children and adolescents about what they have seen, heard, and already know about what is happening regarding the terminally ill family member. Look at how the child/adolescent communicates by paying attention to nonverbal behaviors like body language, facial cues, and tone of voice, all of which will provide cues to the child/adolescent’s comfort level. Additionally, individuals should Listen actively to what the child/adolescent says verbally by paraphrasing their words, asking for clarification, and remaining nonverbally responsive and calm. Children and adolescents deserve to be part of final conversations and have their questions addressed openly in a way that they can understand [21]. There is still much work to be done with regard to children/adolescents and the death and dying process, but open communication is a good place for families to start. 

## 4. Scholarly Challenges and Future Directions

It is critical to discuss the challenges scholars face when conducting research within the final conversations context. To begin, final conversations research is emotionally arousing for participants, as the EOL journey is wrought with novel, challenging emotional experiences. Because of this, we have encountered particular limitations in our research. For instance, our samples are almost always predominantly female, which could be a reflection of societal gender norms that dissuade men from discussing emotionally-charged topics, or could be because women are often tasked to be the primary caregivers at the end-of-life [2,5]. Participants also tend to be predominantly Caucasian, which raises questions about cultural variation in how final conversations manifest within families [22], which is a critical component healthcare providers must consider when handling EOL issues with patients and families. Additionally, from an anecdotal standpoint, we have noticed a tendency for participants’ retrospective accounts of final conversations to be mainly positive [2]. While we have heard some of the final conversations that were challenging and difficult, we are aware that there are many people who choose to avoid the situation completely or who have had such negative final conversations that they may have chosen not to share their experiences with us. There is much more to be learned about this aspect of final conversations. Less homogenous and more representative samples are needed. We call upon researchers with access to understudied populations and contexts to address this salient gap in the literature. 

The familial context, and its influence on final conversations, needs to be further explored [3]. Specifically, scholars could seek to understand how communicative beliefs and norms, as well as relational dynamics within the family influence the frequency and quality of final conversations during the end of life. In addition, researchers might examine how family members (e.g., siblings, spouses) collaboratively narrate their final conversations experiences with a loved one after death, as well as how family relationships outside of the terminally ill person are affected by final conversations. 

Finally, longitudinal work on final conversations would help establish causal links between relational messages exchanged at the end-of-life, their antecedents, and outcomes. In particular, it would be revealing to explore how the frequency and quality of final conversations potentially influences post-death social, psychological and physiological well-being outcomes (e.g., bereavement, grief, personal growth, support network outreach, stress, depression, etc.), and this influence would be best studied via longitudinal analysis. Longitudinal research could also help researchers understand the causal influence of particular antecedents, like family communication norms and beliefs, on the frequency and quality of final conversations during the EOL journey. To move forward, scholars could employ a longitudinal design in three phases: (1) survey members of families at the onset of a family member’s terminal diagnosis in order to assess family communication environments, religious orientation, and relationship quality among family members; (2) survey participants again post-death about their final conversation experiences prior to death, as these memories would be especially poignant; (3) survey participants a final time two to three years later to assess variables related to social, psychological and physical well-being. 

## 5. Conclusions

This manuscript has outlined the findings from the body of research on final conversations. In addition, we have provided healthcare and palliative care professionals, as well as family members, practical implications to assist in the EOL journey. Understanding final conversations and their impact on families is of crucial importance to the academic and professional healthcare communities, as well as to individuals and their families as they face an impending death. Fortunately, the surface of EOL family communication research has been scratched, thanks in part to the dedicated scholars who aim to help individuals through the EOL journey, and also to those who have opened their lives as participants in research. We are confident these findings will be useful to many, but we are also aware that more questions remain unanswered; thus, while we have come a far way, more work is still needed.

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
