# Peer review of "Final Conversations: Overview and Practical Implications for Patients, Families, and Healthcare Workers"

_behavsci, 2017, doi:10.3390/bs7020017_

Round 1
Reviewer 1 Report
I enjoyed reading this manuscript summarizing the body of work on final conversations. This paper has several strengths, including the significance of the topic to researchers and to practitioners. However, there is room for a great deal of improvement in how the content is presented. As a reader, I was actually quite frustrated by the end of the paper because there were so many instances of lack of precision in the writing, omissions of key details, and grammatical errors. If I had not been a reviewer, I would have stopped reading by the third page. The good news is that many of these issues are easily correctable. I hope you find my feedback helpful as you continue your work on this manuscript.
Overall:
I found it very strange to use the abbreviation "FCs" throughout the manuscript. For example, in the abstract, you state "FCs takes the family member's perspective and highlights..." It seems like the sentence should read "FCs TAKE...and HIGHLIGHT" or "AN FC takes...and highlights." Another example in line 248: It makes no sense to talk about "a negative FCs." If you decide to keep the convention of using FCs as a singular noun, please provide a rationale for this, since it is very disruptive to the reader. This is an issue throughout the manuscript, so whatever you decide to do, please make consistent changes throughout the paper. (In fact, the only time I didn't get tripped up by this was in line 105 when you used the singular form "FC," which makes much more intuitive sense.)
I understand that it is grammatically sound to refer to terminally ill people as "the terminally ill." However, I am resistant to this use because it defines the person in terms of the label alone. It is akin to referring to individuals with mental illness as "the mentally ill." I encourage you to instead use the phrase "terminally ill person/people" throughout the manuscript to highlight the personhood of the individuals rather than their medical condition.
There are sloppy mistakes sprinkled throughout the paper. For example: the word "include" (or some synonym) is missing in line 30, "move-on" in line 51 should not be hyphenated, there is a missing comma in line 59 and line 186, there is an error in subject-verb agreement in lines 74-75, there is a missing slash between "and" and"or" in line 97 (and for what it's worth, "and/or" is usually replaceable by just "or" in most cases, which is more precise than "and/or"), there are run-on sentences in line 99 and line 220, lines 194-196 make no grammatical sense.
Summary of FC research
The first statement of this section had me scratching my head. It's like there is a sentence missing right before it that should clarify that most EOL research has focused on the perspective of patients. But instead, you jump in and reference "the other person" but don't give us any clue who this person is "other" than.
You summarize research on the themes and functions of final messages, but you do not include any information about how this data was collected. I surmised from later references in the paper that this research was mostly in-depth interviews, but this is very important information and should be described explicitly. I encourage you to include a paragraph summarizing the specific forms of data collection that you used in this body of work. Was it not quantitative, given that later you say that you wanted to validate these themes quantitatively? Furthermore, when you mention that an additional theme emerged in "this analysis" (line 109), what analysis is that?
Given that you are summarizing this literature, it seems strange to use the past tense (as you might in a results section). Instead, consider using the present tense.
In the paragraph describing the third theme (lines 79-86), you state that spiritual experiences "might have been experienced by one or the other" - who is one or the other? This is an example of a lack of precision in language that leads to confusion on the part of the reader. Moreover, the rest of the sentence doesn't make grammatical sense - who or what "would describe a phenomenon that could not be logically explained"?
The paragraph contained in lines 118-122 sticks out as odd. First, the language here is more colloquial ("That being said," "doesn't") than in other parts of the paper. Second, the content is underdeveloped. I encourage you to either elaborate in greater detail in this paragraph, or cut it altogether.
In line 130 you mention that they "discuss the issue of terminal time" - does that mean the discussion is in previous research or in the current manuscript?
How does time create the framework for the communicative context at the end of life (lines 131-132)? This is an interesting idea, but the lack of elaboration leaves me wondering what is meant here.
I recommend cutting the word "Finally" in line 150.
Rather than saying "for some reason" (which implies that it is irrational behavior) people choose private, I encourage you to re-word the sentence in line 154 to read something like "disclose information to the terminally ill person, but may also feel constrained to keep..."
The discussion of the tension of openness-closedness is very consistent with the concept of "social death," which is commonly cited in the literature. I encourage you to make connections to this literature in your discussion in lines 159-165.
You state that most people are not good at expressing negative emotions (line 170). How so? This paragraph on the tension of expression-concealment would benefit from more elaboration.
Practical Implications
I encourage you reconsider how you have labeled the anchors of the suggested needs assessment (line 209). It seems that using anchors such as "I want to talk about this" and "I do not want to talk about this" would be much more user-friendly. The current anchors that you use are very researcher-friendly, but not as intuitive to people who aren't communication scholars.
The word "often" appears twice in the same sentence in lines 219-221.
When you reference "these interviews," what are you referring to? This is yet another example of how you are assuming a level of knowledge in the reader that most of your readers simply won't have, and it can be very frustrating to try to fill in all the blanks that you have left.
Scholarly Challenges and Future Directions
I agree that it is critical to discuss the practical and ethical challenges facing scholars in conducting EOL research, but we can't assume that every reader will be so amenable to our claim. I encourage you to back this claim up with reasons.
When you state that "this can manifest in myriad ways" in lines 252-253, what is the "this" in that sentence? This is a dangling modifier.
Longitudinal research is a good direction for future work, but can you be more specific about how other scholars can move this program of research forward? What research questions are particularly pressing? What forms of data collection are particularly promising? Is there anything else you can recommend in addition to longitudinal research? This is your chance to really dream big. Help readers think creatively about the possibilities here.
Conclusion
You state that "the current body of literature regarding FCs is instrument in assisting patients and their families [to] have a 'good' death" (lines 270-272, incidentally the word "to" is missing). The manuscript in its current form does not demonstrate this. In fact, this is the first reference to a "good" death in the paper. I encourage you to revise this conclusion to summarize what you actually lay out in this paper (and maybe suggest that future research look at health outcomes of FCs that are specifically related to what health scholars define as a "good" death).
Author Response
We would like to thank the reviewers for taking their time, energy, and effort to review this paper. It is a better paper because of your feedback.
Please see attached file for specific responses.

Reviewer 2 Report
The article mentions the FC scale and it would have been nice to see a copy in an appendix perhaps. Also, no reason not to have even a frequency count of types of statements, after all the authors mention 12 years of data so let's see some. If you include some kind of frequency count, then a sample of each communication type would be expected. More could have been made of the additional theme that emerged (instrumental death talk), which to me was interesting because I did not have the benefit of that with my other, who is now dead.
Just recently I watched an interview with a hospice chaplain on the PBS Newshour and she said many people were fascinated by the topic of last words and her advice to them was similar a message from this article (if you have something to say, don't wait until it is too late to say it).
And, not to be picky but at the beginning I did notice a tendency toward choppy sentences (maybe I stopped noticing). For example 44/45, 73/74 sentences could have flowed better together.
I find this topic fascinating and want to see this published so please consider making those small additions.
Author Response
We would like to thank the reviewers for their thoughtful responses. We appreciate the reviewers for taking their time, energy, and effort to give us their feedback.
Please see the attached document for our detailed responses.

Round 2
Reviewer 1 Report
This paper is much improved! I enjoyed reading this revision. In particular, I appreciated your section on methodology as well as your suggestions for utilizing the FC scale in clinical settings. There are still a number of changes that could make your paper even stronger. I hope you find this feedback helpful as you continue working on this paper.
Conceptual issues
- I realize that the inclusion of percentages of messages for the five themes has been added in this revision, likely at the request of another reviewer. However, this is not appropriate. Your methodology did not include a content analysis of a representative sample of final conversations, and therefore the percentages are not only meaningless, they are misleading. I urge you to remove them from the final draft of the paper.
- In line 129 you say that individuals who have difficult relationship talk "focus solely" on the difficult relationship talk. In other words, these individuals do not talk about any of the other themes? That seems implausible.
- How are you specifically working to revise the operationalization of identity (line 157)? This would be useful information to readers who want to pick up this line of research.
- In line 226, you say that not talking about end-of-life issues results "in a waste of time and energy talking about unimportant and superficial topics." And yet, earlier in the manuscript you described how everyday things can be important and functional topics in end-of-life conversations. Perhaps instead you could say that not talking about end-of-life issues results in "a lost opportunity for meaningful connection" or something along those lines.
In line 239, you say that some families have expectations for not showing weakness. What counts as "weakness" here? Are you referring to showing emotion = weakness? Please clarify.
- You suggest that your samples are predominantly women because of societal gender norms that "dissuade men from discussing emotionally-charged topics" (line 307), which is likely true. However, it could also be the result of societal norms that place responsibility for end-of-life caregiving on women more often than on men.
- I'm curious about why you recommend waiting to talk with family members about their FC experience until 12 months after a patient's death (line 337). That seems pretty late to me. Would interviewing them sooner be beneficial as well? In my experience, I have found that people are often willing to talk about their experience soon after a loved one's death because they welcome the opportunity to process their thoughts and feelings.
Grammatical issues
One of the biggest issues I saw was the abundance of grammatical errors. Here is a list of the ones I caught, but there are likely more to be found. I strongly encourage you to carefully proofread your paper before sending it out again.
- Line 18: "serve" is missing an "s."
- Line 26: "end of life" is missing hyphens.
- Line 42: "focus" seems like it should be in past tense.
- Line 50: "give" should be "gives."
- Line 83: comma is missing after "nears."
- Line 131: "hope" is misspelled.
- Line 161: "infer" should be "imply" (a conclusion cannot technically infer, it can only be inferred).
- Line 170: "All together" should be "altogether."
- Line 190: "like" should be omitted (otherwise, without the first dependent clause, the sentence technically reads "How is the terminally person physically looking like today?" which is grammatical incorrect).
- Line 210: comma is missing after "death."
- Line 253: comma is missing after "conversations."
Stylistic issues
- This paper suffers from an overuse of the "slash" sign (/). I encourage you to be more precise in your writing and not rely so heavily on slashes in your writing. For example, I found myself tripping over the "terminally ill person/people" every time it appeared (which was a lot). Instead, mix it up and sometimes use "person" and sometimes use "people." (This phrase appears in lines 27, 35, 42, 50, 54, 84, 107, 111, 125, 129, 149, 163, 175, 181, 188, 189, 193, 197, 214, 220, 224, 242, 257, 271, 274, 324). In addition, the phrase "and/or" can always be replaced with "or." (This appears in lines 97, 216, 229). Please reconsider your use of the slash in line 298 and in line 332.
Author Response
Comments in response to review for “Final Conversations: Overview and Practical Implications for Patients, Families, and Healthcare Workers” all major changes are highlighted in yellow on the manuscript and are as follows:
Conceptual Points:
Percentages were dropped. Ironically, the authors had already dropped them (after momentarily adding them in response to a 2nd reviewer; but we totally agree that these numbers are meaningless—I inadvertently uploaded the pen-ultimate revision that had the percentages). So thank you for supporting our decision to NOT have the percentages. I am sorry that you had to see the percentages at all.
Line 129. We agree “difficult relationship talk” was probably not the only topic talked about during their final conversations; however it was the topic that the participants focused on and believed that it was the MOST important topic of their final conversations. Thus, wording was changed to: For the individuals that have these challenging relationships, they report that difficult relationship talk was the most important to them during their final conversations.
Line 157. What are we doing to revision the operationalization of identity? Added sentence: Specifically, we returned to the qualitative analyses to create more behaviorally concrete scale items to measure the identity dimension of the Final Conversations Scale.
Line 226. We agree with the reviewer and thank her/him for the suggestion, this new wording is more reflective of the intent of the statement. The end of the sentence now reads “which leads to a missed opportunity for dialogue and connection.”
Line 239. We have clarified statement, it now reads: Finally, some families have expectations that they are supposed to be strong for one another and not display emotions related to sadness, anxiety, and distress.
Line 307. We agree with the reviewer, and we in fact have stated that women are tasked with the duty of being a primary caregiver at the end of life. We added the following to the sentence: or could be because women are often tasked to be the primary caregivers at the end of life.
Line 337. We deleted the following from the sentence “at one to 12 months”, it now reads: 2) survey participants again post-death about their final conversation experiences prior to death. [We agree that talking about the death process often is cathartic for the participants, and in reality we are suggesting that participants be surveyed at 3 different points (before the death, within a brief period of time after the death, and then again 3-4 years later to see the impact of Final Conversations from more than one point in time) and that the author of those studies should make their own decision as to the exact timing and they can offer their rationale for their decisions.]
Grammatical Issues
All of the ones noted were cleaned up and the paper was read through multiple more times to catch any other errors. Concerning end of life versus end-of-life, the author (s) have been beat up by reviewers depending on the journal as to which is correct (the use of hyphens or no use of hyphens); we are happy to use either style and simply want to be consistent. So for this paper we consistently used hyphens throughout for end-of-life.
Stylistic Issues
The 2nd reviewer requested that the authors specifically add person/people, so we did. We agree with this reviewer and have taken your suggestion to alternate between person and people where best appropriate. We did however keep the children/adolescents because the counselors that we dealt with concerning children and adolescents explained the importance to that population (especially the adolescents) that there be a distinction between the two groups and they suggested that we use children/adolescents. In regard to religious/spiritual messages, we have also kept this as is because these are two distinct types of talk that were considered two halves of the same theme.
The authors would like to thank the reviewers for their in-depth review and note to attention. We believe that this is a stronger and cleaner paper because of the feedback that we received.